# Two-Step Elution Recovery of Cyanide Platinum Using Functional Metal Organic Resin

**DOI:** 10.3390/molecules24152779

**Published:** 2019-07-31

**Authors:** Muhan Chen, Qun Ye, Shaosong Jiang, Min Shao, Ci Jin, Zhangjie Huang

**Affiliations:** School of Chemistry Science and Engineering, Yunnan University, Cuihu North Road No. 2, Kunming 650091, China

**Keywords:** metal organic resin, platinum cyanide, recovery

## Abstract

A novel functional ion-exchange/adsorption metal organic resin (MOR), TEBAC-HKUST-1, was prepared and characterized. Ethanedithiol was used as the grafting agent to introduce thiol groups onto HKUST-1, and 4-vinylbenzyl chloride was then grafted onto SH-HKUST-1 using thiol groups. Finally, the quaternary ammonium functional group was immobilized onto the carrier by performing a quaternization reaction. The structure and property of TEBAC-HKUST-1 MOR were characterized by TGA, N_2_ adsorption–desorption, FTIR, SEM, and XRD. TEBAC-HKUST-1 MOR was used to remove metal cyanide complexes from wastewater. The adsorption was rapid, and the metal cyanide complexes including Pt(CN)_4_^2−^, Co(CN)_6_^3−^, Cu(CN)_3_^2−^, and Fe(CN)_6_^3−^ were removed in 30 min. TEBAC-HKUST-1 MOR exhibited a high stability in neutral and weak basic aqueous solutions. Furthermore, Pt(II) could be efficiently recovered through two-step elution. The recovery rate of Pt(II) for five cycles were over 92.0% in the mixture solution containing Pt(CN)_4_^2−^, Co(CN)_6_^3−^, Cu(CN)_3_^2−^, and Fe(CN)_6_^3−^. The kinetic data were best fitted with the pseudo second-order model. Moreover, the isothermal data were best fitted with the Langmuir model. The thermodynamic results show that the adsorption is a spontaneous and exothermic process. TEBAC-HKUST-1 MOR not only exhibited excellent ability for the rapid removal of metal cyanide complexes, but also provided a new idea for the extraction of noble metals from cyanide-contaminated water.

## 1. Introduction

Cyanide is one of the most dangerous contaminants in environment, threatening human health and ecological systems [1]. The World Health Organization (WHO) recommends that the level of cyanide in drinking water should be less than 0.05 mg L^−1^ [2,3]. Pressure cyanidation has been extensively used in extracting noble metals from flotation concentrate in China [4]. A large number of platinum group metals containing Pt, Pd, Rh, and Ir in cyanide effluents should be recycled. Therefore, removal of cyanide and recovery of noble metals from cyanide-contaminated water are important tasks. The main species of cyanide are free cyanide and metal cyanide complexes. In the past decades, various methods have been applied to remove metal cyanide complexes from wastewater, including biological degradation [5,6,7], chemical oxidation [8], and ion-exchange/adsorption [9]. Most metal cyanide complexes exhibit a wide range of biological stability compared to free cyanide ions; hence, metal cyanide complexes cannot be treated by biological degradation [6]. Chemical oxidation often results in undesired byproducts, which can cause secondary pollution to water body. Ion-exchange/adsorption is considered as a relatively low cost and highly efficient method for the treatment of ionic pollutants [10]. However, common ion-exchange organic resins exhibit a relatively slow sorption kinetics, low thermal and chemical stability, moderate selectivity, and poor regeneration and reusability for metal cyanide complexes [11].

Recently, metal organic resins (MORs) with ion-exchange/adsorption properties have attracted much attention owing to their fast ion-exchange kinetics, unique crystalline porous structure, high ion-exchange/adsorption capacity, and high selectivity for toxic ions [12]. MORs are known as the next-generation ion-exchange/adsorption adsorbents [13]. MORs exhibit an excellent ability to remove various hazardous contaminants in water, including Cr(VI) [14], SeO_4_^2−^/SeO_3_^2−^ [15], PO_4_^3−^ [16], F^−^ [17], ClO_4_^−^ [18], NO_3_^−^/NO_2_^−^ [19], As(V)/As(III) [20], Hg^2+^ [21], Pb^2+^ [22] and Cd^2+^ [23].

Because of the tunability of cations and anions, quaternary amine salts have been widely used in the extraction and separation of metal cyanide [24,25,26,27]. However, quaternary ammonium extraction agents have several drawbacks such as easily emulsified and unsuitable viscosity. An effective solution to these problems is the immobilization of quaternary amine salts onto a solid material with hydrophilic–lipophilic matrix [28].

Herein, a novel functional ion-exchange/adsorption metal organic resin (TEBAC-HKUST-1 MOR) was prepared by post-synthetic modification strategy. First, Cu^2+^ ions of HKUST-1 were chelated with the S atom of ethanedithiol. Ethanedithiol was used as the grafting agent to introduce thiol groups onto HKUST-1. Second, the 4-vinylbenzyl chloride was immobilized onto SH-HKUST-1 by the reaction of vinyl and thiol. Finally, quaternary ammonium functional groups were grafted onto the MOF matrix through quaternization reaction. To the best of our knowledge, the use of functional HKUST-1 as ion-exchange/adsorption MOR in the removal of metal cyanide complexes from wastewater has not been reported. The structure and property of TEBAC-HKUST-1 MOR were systematically characterized by thermogravimetric analysis (TGA), N_2_ adsorption–desorption, Fourier transform infrared (FTIR) spectroscopy, scanning electron microscopy (SEM), and X-ray diffraction (XRD).

Because of the combination of merits of quaternary ammonium ion exchange with high porosity of HKUST-1 [29], TEBAC-HKUST-1 MOR exhibited rapid ion-exchange/adsorption performance for Pt(CN)_4_^2−^, Co(CN)_6_^3^^−^, Cu(CN)_3_^2−^, and Fe(CN)_6_^3^^−^, and almost all the metal cyanide complexes could be removed in 30 min. Furthermore, the adsorbed Pt(CN)_4_^2−^ could be selectively recovered by two-step elution. First, the loaded Co(CN)_6_^3^^−^, Cu(CN)_3_^2−^, and Fe(CN)_6_^3^^−^ on TEBAC-HKUST-1 MOR could be eluted using a NaCl solution. Subsequently, a NH_4_SCN solution was used to elute the loaded Pt(CN)_4_^2−^. TEBAC-HKUST-1 MOR exhibited efficient separation for Pt(CN)_4_^2−^ from a mixed metal cyanide complex mixture containing Pt(CN)_4_^2−^, Co(CN)_6_^3−^, Fe(CN)_6_^3−^, and Cu(CN)_3_^2−^, as well as excellent reusability. Adsorption isotherms, kinetics models, and adsorption thermodynamics of Pt(CN)_4_^2−^ were also systematically investigated.

## 2. Results

### 2.1. Characterization

#### 2.1.1. FTIR Spectra

Figure 1 shows the FTIR spectra of HKUST-1, TEBAC-HKUST-1, and TEBAC-HKUST-1-Pt(CN)_4_^2−^. The bands at 1620 and 1438cm^−1^ show the vibrations of carboxylate groups of HKUST-1 while the strong band at 1566 and 1378 cm^−1^ assigned to bending stretching of benzene ring of the HKUST-1 [30]. The bands located at 770 cm^−1^ can be attributed to Cu−O bond (Figure 1a) [31]. These characteristic peaks are consistent with the previously reported FTIR spectrum of HKUST-1. The characteristic vibrational band of C–S group appeared at 686 cm^−1^, indicating that thiol groups were successfully introduced into HKUST-1 matrix [32]. Furthermore, compared with those observed for the bare HKUST-1, some new peaks were observed at 2973, 2916, and 2849 cm^−1^, corresponding to the C–H stretching of alkyl groups [33], indicating that quaternary ammonium was successfully grafted onto SH-HKUST-1 framework through quaternization reaction (Figure 1b). Compared to TEBAC-HKUST-1, the IR spectra of adsorbed species of TEBAC-HKUST-1-Pt(CN)_4_^2−^ did not exhibit any remarkable shift (Figure 1c), while new absorption peaks corresponding to the C≡N stretching vibration of platinum cyanide were observed at 2208 cm^−1^ and 2182 cm^−1^ [34].

#### 2.1.2. XRD Spectra

The XRD patterns of obtained samples HKUST-1, TEBAC-HKUST-1, after five adsorption-desorption cycles TEBAC-HKUST-1 MOF at pH = 8.0, the simulated sample from the single crystal data of HKUST-1, and TEBAC-HKUST-1 MOR at different pH values are shown in Figure 2a, b, c, d and e, respectively. The main characteristic diffraction peaks of the as-synthesized HKUST-1 match well with those of simulated single-crystal structure (CCDC: 112954/www.ccdc.cam.ac.uk). Good crystallinity showed that the as-synthesized sample has a pure phase of HKUST-1. Compared with HKUST-1, the main characteristic diffraction peaks of TEBAC-HKUST-1 showed slight differences with those of HKUST-1, confirming that ligand functionalization does not change the original crystal structure of sample [32]. After five adsorption-desorption cycles at pH=8.0, the intensities of diffraction peaks of TEBAC-HKUST-1 MOR slightly decreased, indicating that the crystallinity of MOR was only partial loss. The TEBAC-HKUST-1 MOR adsorbent exhibited a good stability and reusability.

To evaluate the chemical stability of target MOR, TEBAC-HKUST-1 was first suspended in aqueous solutions at pH = 7.0–12.0, followed by XRD measurements to monitor the changes in the crystallinity of MOF. As shown in Figure 2e, the crystallinity of sample does not show a significant loss at various pH values ranging from 7.0 to 9.7 (room temperature). TEBAC-HKUST-1 showed a high-water stability in neutral and weakly basic aqueous solutions. When the pH of solution reached 11, TEBAC-HKUST-1 MOR partially decomposed.

#### 2.1.3. SEM Analysis

Figure 3a–c shows the SEM images of as-prepared HKUST-1, TEBAC-HKUST-1 MOR, and recovered TEBAC-HKUST-1 MOR, respectively. HKUST-1 particles exhibited a regular octahedron shape with an average particle size of 20~30 μm. The morphology of as-synthesized HKUST-1 samples was consistent with that reported in literature [35]. Compared with that observed for bare HKUST-1, TEBAC-HKUST-1 MOR and the recovered TEBAC-HKUST-1 MOR maintained the same octahedron structure [36], indicating that HKUST-1 functionalization and adsorption reaction did not significantly affect the HKUST-1 structure, consistent with the XRD analyses.

#### 2.1.4. TGA

The TG curves of HKUST-1 and TEBAC-HKUST-1 MOR are shown in Figure 4. The TG curves of HKUST-1 show three weight loss signals at 40–150 °C, 150–300 °C, and 300–360 °C, corresponding to the loss of physically adsorbed water, the desorption of coordinated water with copper ion or crystal water of HKUST-1, and collapse of HKUST-1 framework, respectively [37]. For the TEBAC-HKUST-1 MOR, the weight loss can also be divided into three stages. First, the departure of adsorbed or coordinated water molecules inside the sample (<200 °C). The percent of weight loss of HKUST-1 from 40 °C to 200 °C was more than TEBAC-HKUST-1 MOR, because a part of the sites for water in HKUST-1 framework was replaced with the grafted quaternary ammonium [38]. Second, the weight loss was due to the decomposition of immobilized quaternary ammonium (200–285 °C) [38]. The third weight loss stage above 285 °C was assigned to the collapse of MOR frameworks. TGA confirmed that TEBAC-HKUST-1 MOR has good thermal stability.

#### 2.1.5. N_2_ Adsorption–Desorption Isotherms

Figure 5 shows the N_2_ adsorption–desorption isotherms of as-prepared HKUST-1 and TEBAC-HKUST-1 MOR. The N_2_ adsorption–desorption isotherm of as-synthesized HKUST-1 samples exhibited type-I isotherms (Figure 5a). The as-prepared materials are therefore essentially microporous, consistent with the previously reported N_2_ adsorption–desorption isotherm for HKUST-1 [39]. Compared with bare HKUST-1, the N_2_ adsorption–desorption isotherms of TEBAC-HKUST-1 MOR showed similar type-IV isotherms (Figure 5b). This is probably because HKUST-1 framework was partially decomposed during functionalization, providing some mesopores in the MOR. Similar results were reported by Alavi and co-workers [40]. The surface area and total pore volume of as-prepared HKUST-1 and TEBAC-HKUST-1 MOR (N, 1.5 wt%) are shown in Table 1. Compared with as-prepared HKUST-1, both the BET surface area and total pore volume of TEBAC-HKUST-1 MOR decreased significantly. This is because the pores of as-prepared HKUST-1 framework were partially occupied by functionalized groups. This also indicates that quaternary ammonium was successfully immobilized onto the MOF framework.

#### 2.1.6. XPS

As shown in Figure 6a, C, O, and Cu were mainly observed for HKUST-1 and TEBAC-HKUST-1 because of their skeleton structure according to the wide-scan XPS spectrum. Figure 6a also shows that the XPS spectrum of TEBAC-HKUST-1 MOR contains six elements: Cu, O, C, N, Cl, and S.

Figure 6b shows the sulfur 2p XPS spectrum of TEBAC-HKUST-1 MOR sample. The S 2p peak was resolved into three components; the binding energy (BE) components were observed at 164.8, 163.4, and 162.0 eV, corresponding to H–S, C–S, and Cu–S bonds, respectively [41,42]. Figure 6c shows that only one peak appeared at 401.9 eV in the N 1s XPS high-resolution spectra of TEBAC-HKUST-1 MOR, consistent with those for previously reported quaternary ammonium [43]. This indicates that quaternization reaction occurred, and quaternary ammonium functional groups were grafted onto the MOFs.

By comparing the wide-scan spectra before and after the adsorption of Pt(CN)_4_^2−^, Pt 4f bands were clearly observed in the spectra after the adsorption of Pt(II), indicating that Pt(CN)_4_^2−^ ions were successfully adsorbed on TEBAC-HKUST-1 MOR. After the adsorption, a new peak was observed in the N 1s spectra of TEBAC-HKUST-1-Pt(CN)_4_^2−^ at a BE of 398.6 eV, indicating that CN^−^ is adsorbed on TEBAC-HKUST-1 MOR (Figure 6d) [44]. Furthermore, from the Pt 4f spectra (Figure 6e), Pt was clearly observed, confirming the successful adsorption of Pt(CN)_4_^2−^ on the modified MOFs. The Pt 4f_7/2_ peak at a BE of 73.20 eV (DS = 3.35 eV) corresponded to platinum cyanide groups (Pt(CN)_4_^2−^). By comparing the XPS spectra of pristine Pt(CN)_4_^2−^ with that after the adsorption (Figure 6e), the adsorption of Pt(CN)_4_^2−^ on TEBAC-HKUST-1 MOR was confirmed to exert no effect on Pt 4f spectra, indicating that a coordinate covalent bond was not formed between Pt(CN)_4_^2−^ and TEBAC-HKUST-1 MOR. Therefore, the XPS measurements further supported that ion-exchange mechanisms are possibly the major adsorption mechanisms. The results are in consistent with those obtained from the FTIR spectra.

### 2.2. Effects of pH

The effects of pH on the adsorption of metal cyanide complexes were evaluated using single-component solutions at 25 °C. The experimental parameters were fixed as follows: 10 mg of TEBAC-HKUST-1 MOR, adsorption time of 30 min, initial Pt(II), Fe(III), Cu(I), or Co(III) concentration, 50.0 mgL^−1^; solution volume, 20 mL. The batch system was used for evaluating the effects of pH on adsorption. Equilibrium loadings of Pt(II), Fe(III), Cu(I), or Co(III) were examined at various pH values ranging from 7.0 to 10.0. The results show that with an increase in the pH from 7.0 to 8.5, the equilibrium adsorption capacities (q_e_) of Pt(II), Fe(III), Cu(I), or Co(III) remained almost constant, and with further increase in solution pH, the q_e_ values significantly decreased. At a higher pH, OH^−^ ions are abundant in solution, thus making OH^−^ ions compete with Pt(CN)_4_^2−^, Co(CN)_6_^3^^−^, Cu(CN)_3_^2−^, and Fe(CN)_6_^3^^−^, causing a decrease in the q_e_ values of metal cyanide complexes [45]. The experimental results indicate that the main mechanism for the adsorption of metal cyanide complexes from aqueous solutions followed an ion-exchange reaction:nM – R_3_N^+^ Cl^−^ (S) + Me(CN) _m_^n−^(aq)= (M-R_3_N)_n_^+^ Me(CN)_m_^n−^ (s) + nCl^−^(aq)(1)

Here, M denotes the MOR matrix. Finally, pH 8 was selected for the subsequent experiments.

### 2.3. Maximum Adsorption Capacities

The batch system was used for the removal of metal cyanide complexes from aqueous solutions. The experimental maximum sorbent capacity was obtained according to previously reported method. The obtained maximum sorbent capacity of TEBAC-HKUST-1 MOR (N, 1.5 wt%) was also compared with AC and as-prepared HKUST-1. The experimental data are shown in Figure 7. As shown in Figure 7, TEBAC-HKUST-1 MOR exhibited excellent adsorption performance towards Pt(CN)_4_^2−^, Co(CN)_6_^3−^, Cu(CN)_3_^2−^, and Fe(CN)_6_^3−^ compared with AC and as-prepared HKUST-1, indicating that the quaternary ammonium group plays a key role in the removal of metal cyanide complexes.

### 2.4. Adsorption Kinetics

To evaluate the kinetic parameters of Pt(CN)_4_^2−^ on TEBAC-HKUST-1 MOR(N, 1.5 wt%), pseudo-first order, pseudo-second order, and intraparticle models were used to fit the experimental data. The three models can be expressed as Equations (2–4), respectively:lg(q_e_ − q_t_) = lgq_e_ − k_1_t/2.303,(2)
t/q_t_ = 1/(k_2_ q_e_^2^) + t/q_e_,(3)
q_t_ = k_p_ t^1/2^ + C,(4)

Here, q_e_ and q_t_ are the amount of loading of Pt(II) at equilibrium and at any time (mg g^−1^), respectively; k_1_ is the rate constant of pseudo-first-order kinetics (min^−1^); k_2_ is the pseudo-second-order constant (g mg^−1^ min^−1^); k_p_ is the intraparticle diffusion rate constant (mg g^−1^ min^−0.5^); C is the boundary layer thickness. The experimental conditions were as follows: 10 mg of MOR; equilibrium time (t < 30 min); initial Pt(II) concentration, 100 mg L^−^^1^; pH = 8.0, solution volume, 20 mL. The parameters for the three kinetic models are shown in Table 2.

As shown in Table 2, the adsorption kinetics for Pt(II) well fitted with the pseudo-second-order kinetic model. Compared with polymer resin, the adsorption of Pt(CN)_4_^2−^ using TEBAC-HKUST-1 MOR was more rapid; the equilibrium was established within 30 min (Figure 8).

Because of unique tunnels and crystalline porous structure of TEBAC-HKUST-1 MOR, Pt(CN)_4_^2−^ could rapidly spread into the MOR matrix. In contrast, a hydrophobic polymer resin and AC, TEBAC-HKUST-1 showed a quick adsorption equilibrium for Pt(CN)_4_^2−^. More than 2 h was taken when Pt(CN)_4_^2−^ was adsorbed on the polymer resin or AC.

### 2.5. Sorption Isotherms

Isotherm models show how metal cyanide complexes are distributed between the solution and TEBAC-HKUST-1(N, 1.5 wt%). In this study, Langmuir and Freundlich isotherm models were used for fitting the experimental data. The experimental parameters were as follows: 10 mg of MOR; t = 30 min; pH = 8.0; initial Pt(II) concentration, 10–200 mg L^−1^; solution volume, 20 mL. The Langmuir and Freundlich equilibrium models are as follows:C_e_/q_e_= 1/(q_m_ b)+C_e_/q_m_,(5)
lgq_e_ = lgK_F_ + n^−1^ lgC_e_,(6)

Here, q_m,_ b, K_F,_ and 1/n are the maximum adsorption capacity (mg g^−1^), Langmuir adsorption equilibrium constant (L mg^−1^), Freundlich constant (L g^−1^), and adsorption intensity, respectively. The parameters for the two isotherm models are shown in Table 3 and Figure 9.

As shown in Table 3 and Figure 9, the adsorption of Pt(CN)_4_^2−^ on TEBAC-HKUST-1 MOR well conforms to the Langmuir equation. The q_m_ value was calculated from the Langmuir equation to be 289.9 mg g^−1^, slightly lower than the maximum experimental adsorption capacity of 290.2 mg g^−1^. Moreover, TEBAC-HKUST-1 MOR exhibited a higher maximum adsorption capacity than other sorbents reported earlier (Table 4).

### 2.6. Thermodynamic Parameters

The thermodynamic parameters including ∆H, ∆G, and ∆S were measured according to the following Equation:∆G = ∆H – T ∆S,(7)
K_C_ = (C_0_-C_e_) V/(M C_e_),(8)
lnK_C_ = ∆H/(R T) + ∆S/R,(9)
where Kc, C_o_, C_e_, V, and M are the equilibrium constant, initial concentration, equilibrium concentration, volume of Pt(CN)_4_^2−^ solution, and mass of TEBAC-HKUST-1 MOR, respectively. The values of ∆H and ∆S can be obtained from the linearized plot of lnKc versus T^−1^ (Figure 10). The thermodynamic parameters for the absorption of Pt(CN)_4_^2−^ are shown in Table 5. A negative value of ∆H suggests exothermic reaction. Negative ∆S indicates a decreased randomness at the two-phase interface during the adsorption of Pt(CN)_4_^2−^ on TEBAC-HKUST-1 MOR [49,50]. A negative value of ∆G confirmed that the reaction was spontaneous. The values of ∆G increases with increasing temperature, indicating that the adsorption is more spontaneous at lower temperatures.

### 2.7. Removal of Metal Cyanide Complexes and Recovery of Pt(II)

TEBAC-HKUST-1 MOR was used for the removal of metal cyanide complexes and recovery of Pt(CN)_4_^2^^−^from a mixture. Typically, 100 mg of TEBAC-HKUST-1 MOR(N, 1.5 wt%) was added to a 100 mL mixture containing Pt(CN)_4_^2−^, Co(CN)_6_^3−^, Fe(CN)_6_^3−^, and Cu(CN)_3_^2−^. Batch adsorption experiments were carried out under optimum conditions. The adsorption rate of all four metal cyanide complexes was over 99.0% in the mixture.

Furthermore, the adsorbed Co(CN)_6_^3^^−^, Cu(CN)_3_^2−^, and Fe(CN)_6_^3^^−^ could be selectively eluted by 1.5 mol L^−1^ NaCl solution, whereas the elution percentage of Pt(CN)_4_^2−^ was less than 1.0% [51]. Finally, loaded Pt(CN)_4_^2−^ could be eluted using a 2.0 mol L^−1^ NH_4_SCN solution. The recovery rate of Pt(CN)_4_^2−^ was over 97.0%. The adsorbent can be regenerated by washing with a saturated sodium chloride solution. The experimental results show that TEBAC-HKUST-1 MOR could not only be used to efficiently remove metal cyanide complexes, but also could be used to selectively recycle Pt from mixed metal cyanide complexes.

The charge density of Pt(CN)_4_^2−^ is less than those of Cu(CN)_3_^2−^, Co(CN)_6_^3−^, and Fe(CN)_6_^3−^. Fewer water molecules are required to stabilize Pt(CN)_4_^2−^ compared to Co(CN)_6_^3−^, Fe(CN)_6_^3−^, and Cu(CN)_3_^2−^ anions in the aqueous solution. Based on the principle of minimum charge density, Pt(CN)_4_^2−^ should exhibit a higher affinity with quaternary ammonium cations compared to Cu(CN)_3_^2−^, Co(CN)_6_^3−^, or Fe(CN)_6_^3−^. Therefore, Cu(CN)_3_^2−^, Co(CN)_6_^3−^, and Fe(CN)_6_^3−^ adsorbed on TEBAC-HKUST-1 MOR could be eluted more easily compared to Pt(CN)_4_^2−^. Hence, Cu(CN)_3_^2−^, Co(CN)_6_^3−^, and Fe(CN)_6_^3−^ can be preferentially exchanged with Cl^−^ anions. Based on the “perchlorate effect,” the size of SCN^−^ is greater than that of Cl^−^, leading to charge density of SCN^−^ lower than that of Cl^−^. Therefore, the interaction of SCN^−^ with M – R_3_N^+^ is considerably stronger than that of Cl^−^, consistent with the experimental results. The adsorbed Pt(CN)_4_^2−^ on TEBAC-HKUST-1 MOR could be completely eluted with SCN^−^. The elution reaction for SCN^−^ ion might involve ion exchange:(M − R_3_N))_2_^+^Pt(CN)_4_^2−^(s) + 2SCN^−^(aq) = 2M – R_3_N^+^SCN^−^(s) + Pt(CN)_4_^2−^(aq),(10)
where M denotes TEBAC-HKUST-1 MOR matrix.

### 2.8. Regeneration Experiment

To investigate the regeneration ability of TEBAC-HKUST-1(N, 1.5 wt%), the recovery rates of Pt(CN)_4_^2−^ were estimated for five adsorption–desorption cycles from a mixed metal cyanide complex solution. The results are shown in Figure 11. As shown in Figure 11, the recovery rates of Pt(CN)_4_^2−^ for all five cycles were over 92.0% in the mixture. According to the experimental results, TEBAC-HKUST-1 MOR(N, 1.5 wt%) exhibited an efficient separation for Pt(CN)_4_^2−^ from a mixed metal cyanide complex solution containing Pt(CN)_4_^2−^, Co(CN)_6_^3−^, Fe(CN)_6_^3−^, and Cu(CN)_3_^2−^, as well as excellent reusability.

## 3. Materials and Methods

### 3.1. Materials and Reagents

Triethylamine, activated carbon (AC), 4-vinylbenzyl chloride (pCMS), and ethanedithiol were purchased from Alfa Aesar (China). Cu(NO_3_)_2_·3H_2_O, benzene tricarboxylic acid (BTC), 2,2′-azobis(2-methylpropionitrile) (AIBN), K_2_Pt(CN)_4_, K_3_Co(CN)_6_, CuCN, and K_3_Fe(CN)_6_ were purchased from Sigma-Aldrich. All other reagents used in this study were commercially available analytical-grade reagents. Cu(CN)_3_^2−^ was prepared according to literature report [11]. The metal salts were mixed together in water as needed.

### 3.2. Apparatus

Pt(II), Co(III), Cu(I), and Fe(III) concentrations were determined using an ICP-AES instrument (ICAP 6300, Thermo Fisher Scientific, Waltham, MA, USA). The surface area of the adsorbent was measured using a Micromeritics Tristar apparatus (Micromeritics Instrument Corporation, Norcross, GA, USA). Sample morphologies were observed by SEM (FEI Nova NanoSEM 450, Hillsboro, OR, USA). FTIR spectra (400–4000 cm^−1^) were recorded using a Thermo NICOLET 8700 spectrometer (Thermo Fisher Scientific, Waltham, MA, USA). Thermal properties of samples were investigated by TGA (8–800 °C) under nitrogen using SDT-Q600, USA. The thermogravimetric (TG) curves of samples were obtained using a TG analyzer (TGA, SDT-Q600, TA Instruments, New Castle, DE, USA) at 25–800 °C under nitrogen. The phase structure and composition of samples were determined by XRD (Shimadzu, Japan) in the 2θ range from 5° to 50°. Elemental analysis of TEBAC-HKUST-1 MOR was obtained from an Elementar Vario EL III (Elementar, Langenselbold, Germany).

### 3.3. Preparation of TEBAC-HKUST-1 MOR

Scheme 1 shows the three-step preparation of target MOR. First, HKUST-1 was obtained using a solvothermal method following the literature reports [35]. Subsequently, SH-HKUST-1 was prepared as follows [36]: 1.0 g HKUST-1 and 0.25 g ethanedithiol were added into 100 mL of anhydrous toluene. The reaction mixture was continuously stirred for 24 h at room temperature. SH-HKUST-1 was washed with absolute ethanol and dried in a vacuum oven at 40 °C for 8 h. Finally, 1.00 g of SH-HKUST-1 was added into a mixture of 30 mL absolute ethanol, 0.01 g PVP, and 0.01 g AIBN. The reaction mixture was continuously stirred for 8 h at 80 °C. Next, 1 mL of pCMS solution was added into the mixture. The resulting mixture was heated at 80 °C for 24 h [52,53]. Then, 5 mL of triethylamine was added to the mixture and stirred continuously at 85 °C for 4 h [54,55]. The product was collected by centrifugation and washed with methyl benzene. The target MOR was dried in a vacuum oven at 60 °C for 12 h. The elemental analysis results revealed that nitrogen element content in TEBAC-HKUST-1 MOR was 1.5 wt% [56].

## 4. Conclusions

A novel functional ion-exchange/adsorption metal organic resin (TEBAC-HKUST-1 MOR) was prepared following a three-step reaction. Owing to the combination of merits of quaternary ammonium ion exchange with the high porosity of matrix structure of HKUST-1 MOFs, TEBAC-HKUST-1 MOR exhibited rapid ion-exchange/adsorption performance for Pt(CN)_4_^2−^, Co(CN)_6_^3^^−^, Cu(CN)_3_^2−^, and Fe(CN)_6_^3^^−^. The maximum experimental adsorption capacities of Pt (II), Co(III), Cu(I), and Fe(III) reached 290.0, 101.1, 87.3, and 109.2, respectively. Furthermore, the adsorbed Pt(CN)_4_^2−^ could be selectively recovered by two-step elution. First, the loaded Co(CN)_6_^3^^−^, Cu(CN)_3_^2−^, and Fe(CN)_6_^3^^−^ on TEBAC-HKUST-1 MOR could be eluted using a NaCl solution. Subsequently, NH_4_SCN solution was used to elute the loaded Pt(CN)_4_^2−^. TEBAC-HKUST-1 MOR exhibited efficient separation for Pt(CN)_4_^2−^ from a mixed metal cyanide complex solution containing Pt(CN)_4_^2−^, Co(CN)_6_^3−^, Fe(CN)_6_^3−^, and Cu(CN)_3_^2−^, as well as excellent reusability. Adsorption isotherms, kinetics models, and adsorption thermodynamics of Pt(CN)_4_^2−^ on TEBAC-HKUST-1 MOR were also systematically investigated. TEBAC-HKUST-1 MOR not only exhibited excellent ability for the rapid removal of metal cyanide complexes, but also provided a new idea for the extraction of noble metals from cyanide-contaminated water.

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
