# Peer review of "Two-Step Elution Recovery of Cyanide Platinum Using Functional Metal Organic Resin"

_molecules, 2019, doi:10.3390/molecules24152779_

Round 1
Reviewer 1 Report
This work, in general, is well done and written. The presented MOR has shown good effects on removing metal cyanide complexes from water. However, there are still rooms for improvement listed as below.
The PXRD of recovered MOR needs to be included as well.
The authors should try to quantify TEBAC functional groups crafted in TEBAC-HKUST-1. A simple digestion of the MOR followed by HNMR should be able to provide the information. The loading of TEBAC within the MOR can be further correlated to the absorbed metal cyanide to determine whether all active sites have captured metal cyanide.
Scheme 1 needs more work to well present the PSM on the MOR. The authors could cut a unit cell and show the details of the binding between TEBAC and the SBU of HKUST-1. A better figure can help authors quickly understand the material.
I will suggest this work to be accepted for publication if the above mentioned points can be addressed.
Author Response
27-July-2019
27-July-2019
Manuscript ID: molecules-557656
Dear Editors,
Many thanks for the comments on our manuscript, which is so helpful for us to make the manuscript much clearer. We are pleased to answer these questions. The manuscript has been extensively revised according to the editor and the reviewer’s comments. We really appreciate you for your suggestions. We have modified the manuscript accordingly, and detailed corrections are listed below point by point:
According to Reviewer #1’s comment, the revisions were made as follows:
1) The PXRD of recovered MOR needs to be included as well.
Answer:
Thanks for the benefit advice from the reviewer. According to the comment, we have added the PXRD of recovered MOR. After five adsorption-desorption cycles at pH=8.0, the intensities of diffraction peaks of TEBAC-HKUST-1 MOR slightly decreased, indicating that the crystallinity of MOR was only partial loss. The TEBAC-HKUST-1 MOR adsorbent exhibited a good stability and reusability. ( Please see the revised manuscript, Figure 2)
2) The authors should try to quantify TEBAC functional groups crafted in TEBAC-HKUST-1. A simple digestion of the MOR followed by HNMR should be able to provide the information. The loading of TEBAC within the MOR can be further correlated to the absorbed metal cyanide to determine whether all active sites have captured metal cyanide.
Answer:
Thanks for the benefit advice from the reviewer. To date, quantitative calculation of PSM on the MOR through 1H NMR for the one-step synthesis has been widely reported in literature (without complex intermediates) [1-4]. In this study, TEBAC-HKUST-1 MOR main are used in extraction metal cyanide complexes. Hence, the purity of the product is not main goal. Due to grafting reaction from HKUST-1 to TEBAC-HKUST-1 MOR contain three steps, implying the products are complex, containing the raw material and complex intermediates. Hence, purification of TEBAC-HKUST-1 MOR digesting solutions is difficult. The intermediate product will appear vibration peaks in 1HNMR, which serious interference quantify TEBAC functional groups crafted in TEBAC-HKUST-1. Hence, Post-synthetic modification of MOR using multistep functionalization is usually not characterized with 1H NMR[5-7]. The degree of modification usually was calculated based on results of elemental analysis[8]. In this revised manuscript, the content of nitrogen in TEBAC-HKUST-1 MOR was determined through elemental analysis (1.5 wt %.). Based on the elemental analysis result, the grafting rate of active group can be further calculated [8, 9].
As shown in Figure 7, TEBAC-HKUST-1 MOR exhibited excellent adsorption performance towards Pt(CN)42-, Co(CN)63-, Cu(CN)32-, and Fe(CN)63- compared with AC and as-prepared HKUST-1, indicating that the quaternary ammonium group plays a key role in the removal of metal cyanide complexes. From equation (1), we can deduce that the main mechanism for the adsorption of metal cyanide complexes from aqueous solutions followed an ion-exchange reaction. Quaternary ammonium groups provided large amount of active sites, which anionic on quaternary ammonium groups can exchange with Pt(CN)42-, Co(CN)63-, Cu(CN)32-, and Fe(CN)63-. The detailed adsorption mechanism and whether all active sites have captured metal cyanide need to be further studied.
References:
1. Wang, Z.Q.; M. Cohen, S. Postsynthetic Covalent Modification of a Neutral Metal-Organic Framework. J. Am. Chem. Soc. 2007, 129, 12368-12369.
2. Cohen, S.M.Postsynthetic Methods for the Functionalization of Metal Organic Frameworks. Chem. Rev. 2012, 112, 970–1000.
3. Luan, Y.; Zheng, N.N.; Qi, Y.; Yu, J.; Wang, G. Development of a SO3H-Functionalized UiO- 66 Metal–Organic Framework by Postsynthetic Modification and Studies of Its Catalytic Activities. Eur. J. Inorg. Chem. 2014, 26, 4268–4272.
4. Molavi, H.; Eskandari, A.; Shojaei, A.; Mousavi, S.A. Enhancing CO2/N2 adsorption selectivity via post-synthetic modification of NH2-UiO-66(Zr). Microporous and Mesoporous Materials, 2018, 257, 193-201.
5. Chen, C.; Wu, Z. W.; Que, Y. G.; Li, B. X.; Guo, Q. R.; Li, Z.; Wang, L.; Wan, H.; Guan, G. F. Immobilization of a thiol-functionalized ionic liquid onto HKUST-1 through thiol compounds as the chemical bridge. Rsc. Adv. 2016, 6, 54119-54128.
6. Abednatanzi,S.; Abbasi,A.; Masteri-Farahani,M. Post-synthetic modification of nanoporous Cu3(BTC)2 metal-organic. Journal of Molecular Catalysis A: Chemical, 2015, 399,10-17.
7. Haque,E.;Lee,J.E.; Jang,I.T.; Hwang, Y.K.; Chang,J.S. Adsorptive removal of methyl orange from aqueous solution with metal-organic frameworks, porous chromium-benzenedi carboxylates. Journal of Hazardous Materials, 2010, 181,535-542.
8. Chen,J.Z.; Li,K.G.; Chen,L.M.; Liu,R.L.; Huang,X.; Ye.D.Q. Conversion of fructose into 5-hydroxymethyl furfural catalyzed by recyclable sulfonic acid-functionalized metal–organic frameworks. Green Chemistry, 2014,16,2490-2499.
9. Goesten,M.G.; Juan-Alcaniz,J.; Ramos-Fernandez,E.V, et al. Sulfation of metal–organic frameworks: Opportunities for acid catalysis and proton conductivity. Journal of Catalysis ,2011, 281,177-187.
3) Scheme 1 needs more work to well present the PSM on the MOR. The authors could cut a unit cell and show the details of the binding between TEBAC and the SBU of HKUST-1. A better figure can help authors quickly understand the material.
Answer:
We are very sorry that we did not make Scheme 1 clear in the manuscript. According to the comment, we re-state Scheme 1 ( Please see the revised manuscript, Scheme 1).
In this study, a novel functional ion-exchange/adsorption metal organic resin (TEBAC-HKUST-1 MOR) was prepared by post-synthetic modification strategy. First, Cu2+ ions of HKUST-1 were chelated with the S atom of ethanedithiol. Ethanedithiol was used as the grafting agent to introduce thiol groups onto HKUST-1[1,2]. Second, the 4-vinylbenzyl chloride was immobilized onto SH-HKUST-1 by the reaction of vinyl and thiol. Finally[1], quaternary ammonium functional groups were grafted onto the MOF matrix through quaternization reaction[3].
References:
1. Chen, C.; Wu, Z. W.; Que, Y. G.; Li, B. X.; Guo, Q. R.; Li, Z.; Wang, L.; Wan, H.; Guan, G. F. Immobilization of a thiol-functionalized ionic liquid onto HKUST-1 through thiol compounds as the chemical bridge. Rsc. Adv. 2016, 6, 54119-54128.
2. Ke, F.; Qiu, L. G.; Yuan, Y. P.; Peng, F. M.; Jiang, X.; Xie, A. J.; Shen, Y. H.; Zhu, J. F. Thiol-functionalization of metal-organic framework by a facile coordination-based postsynthetic strategy and enhanced removal of Hg2+ from water. J. Hazard. Mater. 2011, 196, 36-43.
3. Wang, M. L.; Hsieh, Y. M. Kinetic study of dichlorocyclopropanation of 4-vinyl- 1-cyclohexene by a novel multisite phase transfer catalyst. J. Mol. Catal. A- Chem. 2004, 210, 59-68.
According to Reviewer #2’s comment, the revisions were made as follows:
1) In this work, ion-exchange resin, and its functionalized form were used, however, no structure was shown anywhere. So, I cannot understand what happens in the resin. Authors should provide structure of resin, and how it was modified.
Answer:
Thanks for the benefit advice from the reviewer. According to the comment, we re-state Scheme 1. ( Please see the revised manuscript).Post-synthetic modification (PSM) refers to the chemical modification of MOR after the formation of crystals, while the basic frameworks remain unchanged. In this study, HKUST-1 is selected as the carrier. Considering the strong coordination ability of metal centers (Cu2+), ethanedithiol is used as the grafting agent to introduce thiol groups. By the reaction of vinyl and thiol, 4-vinylbenzyl chloride can then be immobilized onto HKUST-1, Finally, Quaternary ammonium was grafted onto the MOR for constructing a new adsorbent for the highly effective removal of metal cyanide complexes from aqueous solutions. From Equation (1), we can deduce that the main mechanism for the adsorption of metal cyanide complexes from aqueous solutions followed an ion-exchange reaction.
To date, structural characterization of the chemical modification of MOR through 1H NMR for the one-step synthesis has been widely reported in literature (without complex intermediates) [1-4]. Due to modification reaction from HKUST-1 to TEBAC- HKUST-1 MOR contain three steps, implying the products are complex, containing the raw material and complex intermediates. Hence, purification of the MOR digesting solutions is difficult. The intermediate product will appear vibration peaks in 1HNMR, which serious interference the molecular structural characterization of the sample. Usually, the structure and property of multistep functionalization MOR are systematically characterized by the elemental analysis, FT-IR, XPS, TG , SEM, and N2 adsorption-desorption[5-7] . Furthermore, thiol-functionalization of HKUST-1(the first step)[5,8], the reaction between vinyl and thiol(the second step)[5], and quaterisation reaction(the third step)[9] have been extensively reported in the literatures. Guan et al. used ethanedithiol as grafting agent to introduce thiol groups on HKUST-1. By thiol compounds as the chemical bridge, the ionic liquid then be immobilized onto HKUST-1[5]. Qiu and co-workers illustration of the thiol-functionalization of HKUST-1 through coordination bonding between thiol and Cu2+ in MOR[8]. Quaterisation reaction between triethylamine and benzyl chloride had been reported by Wang and co-workers [9]. Hence, each step synthesis can be contrasted with the literature. The structure and property of the sample have been systematically characterized by following:
(1) The elemental analysis
According to the comment, we have added elemental analysis based on the literature method [10,11]. The elemental analysis results showed that nitrogen element content in TEBAC-HKUST-1 MOR was 1.5 wt %, The elemental analysis result further confirmed the successful grafting of the quaternary ammonium group in the TEBAC-HKUST-1 MOR. ( Please see the revised manuscript).
(2) FTIR spectra
The characteristic vibrational band of C–S group appeared at 686 cm−1, indicating that thiol groups were successfully introduced into HKUST-1 matrix [8]. Furthermore, compared with those observed for the bare HKUST-1, some new peaks were observed at 2973, 2916, and 2849 cm−1, corresponding to the C–H stretching of alkyl groups [12], indicating that quaternary ammonium was successfully grafted onto SH-HKUST-1 framework through quaternization reaction (Figure 1b). ( Please see original manuscript).
(3) XRD spectra
Compared with HKUST-1, the main characteristic diffraction peaks of TEBAC-HKUST-1 showed slight differences with those of HKUST-1, confirming that ligand functionalization does not change the original crystal structure of sample [8]. After five adsorption-desorption cycles at pH=8.0, the intensities of diffraction peaks of TEBAC-HKUST-1 MOR slightly decreased, indicating that the crystallinity of MOR was only partial loss. The TEBAC-HKUST-1 MOR adsorbent exhibited a good stability and reusability. ( Please see the revised manuscript).
(4) SEM analysis
Compared with that observed for bare HKUST-1, TEBAC-HKUST-1 MOR and the recovered TEBAC- HKUST-1 MOR maintained the same octahedron structure [5], indicating that HKUST-1 functionalization and adsorption reaction did not significantly affect the HKUST-1 structure, consistent with the XRD analyses. ( Please see original manuscript).
(5) TGA
For the TEBAC-HKUST-1 MOR, the weight loss can also be divided into three stages. The percent of weight loss of HKUST-1 from 40 °C to 200 °C was more than TEBAC-HKUST-1 MOR, because a part of the sites for water in HKUST-1 framework was replaced with the grafted quaternary ammonium. Second, the weight loss was due to the decomposition of immobilized quaternary ammonium (200 °C to 285 °C) [13] ( Please see original manuscript).
(6) N2 adsorption–desorption isotherms
Compared with as-prepared HKUST-1, both the BET surface area and total pore volume of TEBAC-HKUST-1 MOR decreased significantly. This is because the pores of as-prepared HKUST-1 framework were partially occupied by functionalized groups[8]. This also indicates that quaternary ammonium was successfully immobilized onto the MOF framework. ( Please see original manuscript)
(6) XPS
Figure 6b shows the sulfur 2p XPS spectrum of TEBAC-HKUST-1 MOR sample. The S 2p peak was resolved into three components; the binding energy (BE) components were observed at 164.8, 163.4, and 162.0 eV, corresponding to H–S, C–S, and Cu–S bonds, respectively [14, 15]. Figure 6c shows that only one peak appeared at 401.9 eV in the N 1s XPS high-resolution spectra of TEBAC-HKUST-1 MOR, consistent with those for previously reported quaternary ammonium [16]. This indicates that quaternization reaction occurred, and quaternary ammonium functional groups were grafted onto the MOFs. ( Please see original manuscript)
Hence, the structure and property of TEBAC-HKUST-1 MOR have been systematically characterized by the elemental analysis, FT-IR, XPS, TG , SEM, and N2 adsorption-desorption[5-11].
References:
1. Wang, Z.Q.; M. Cohen, S. Postsynthetic Covalent Modification of a Neutral Metal-Organic Framework. J. Am. Chem. Soc. 2007, 129, 12368-12369.
2. Cohen, S.M.Postsynthetic Methods for the Functionalization of Metal Organic Frameworks. Chem. Rev. 2012, 112, 970–1000.
3. Luan, Y.; Zheng, N.N.; Qi, Y.; Yu, J.; Wang, G. Development of a SO3H-Functionalized UiO- 66 Metal–Organic Framework by Postsynthetic Modification and Studies of Its Catalytic Activities. Eur. J. Inorg. Chem. 2014, 26, 4268–4272.
4. Molavi, H.; Eskandari, A.; Shojaei, A.; Mousavi, S.A. Enhancing CO2/N2
adsorption selectivity via post-synthetic modification of NH2-UiO-66(Zr).
Microporous and Mesoporous Materials, 2018, 257, 193-201.
5. Chen, C.; Wu, Z. W.; Que, Y. G.; Li, B. X.; Guo, Q. R.; Li, Z.; Wang, L.; Wan, H.; Guan, G. F. Immobilization of a thiol-functionalized ionic liquid onto HKUST-1 through thiol compounds as the chemical bridge. Rsc. Adv. 2016, 6, 54119-54128.
6. Abednatanzi,S.; Abbasi,A.; Masteri-Farahani,M. Post-synthetic modification of nanoporous Cu3(BTC)2 metal-organic. Journal of Molecular Catalysis A: Chemical, 2015, 399,10-17.
7. Haque, E.; Lee,J.E.; Jang,I.T.; Hwang, Y.K.; Chang,J.S. Adsorptive removal of
methyl orange from aqueous solution with metal-organic frameworks, porous
chromium-benzenedi carboxylates. J. Hazard. Mater. 2010, 181, 535- 542.
8. Ke, F.; Qiu, L. G.; Yuan, Y. P.; Peng, F. M.; Jiang, X.; Xie, A. J.; Shen, Y. H.; Zhu, J. F. Thiol-functionalization of metal-organic framework by a facile coordination-based postsynthetic strategy and enhanced removal of Hg2+ from water. J. Hazard. Mater. 2011, 196, 36-43.
9. Wang, M. L.; Hsieh, Y. M. Kinetic study of dichlorocyclopropanation of 4-vinyl- 1-cyclohexene by a novel multisite phase transfer catalyst. J. Mol. Catal. A- Chem. 2004, 210, 59-68.
10. Chen,J.Z.; Li,K.G.; Chen,L.M.; Liu,R.L.; Huang,X.; Ye.D.Q. Conversion of fructose into 5-hydroxymethyl furfural catalyzed by recyclable sulfonic acid-functionalized metal–organic frameworks. Green Chem. 2014,16, 2490-2499.
11. Goesten,M.G.; Juan-Alcaniz,J.; Ramos-Fernandez,E.V, et al. Sulfation of metal–organic frameworks: Opportunities for acid catalysis and proton conductivity. Journal of Catalysis, 2011, 281,177-187.
12. Alemdar, A.; Atici, O.; Güngör, N. The influence of cationic surfactants on rheological properties of bentonite-water systems. Mater Lett. 2000, 43, 57-61.
13. Abid, H. R.; Pham, G. H.; Ang, H. M.; Tade, M. O.; Wang, S. B. Adsorption of CH4 and CO2 on Zr-metal organic frameworks. J. Colloid Inter. Sci. 2012, 366, 120-124.
14. Laiho, T.; Leiro, J. A.; Heinonen, M. H.; Mattila, S. S.; Lukkari, J. Photoelectron spectroscopy study of irradiation damage and metal-sulfur bonds of thiol on silver and copper surfaces. J. Electron Spectrosc. 2005, 142, 105-112.
15. Muench, F.; Fuchs, A.; Mankel, E.; Rauber, M.; Lauterbach, S.; Kleebe, H. J.; Ensinger, W. Synthesis of nanoparticle/ligand composite thin films by sequential ligand self assembly and surface complex reduction. J. Colloid Interf. Sci. 2013, 389, 23-30.
16. Cao, W.; Wang, Z. Q.; Zeng, Q. L.; Shen, C. H. 13C NMR and XPS characterization of anion adsorbent with quaternary ammonium groups prepared from rice straw, corn stalk and sugarcane bagasse. Appl. Surf. Sci. 2016, 389, 404-410.
2) Authors explained the higher selectivity of Pt by different affinity with quartenary ammonium ion. However, resins that was not grafted with ammonium ion, also exhibited selectivity for Pt. Why?
Answer:
We are very sorry that we did not state clearly about Fig.7. We added Y-axis units of Fig.7(mg/g)( Please see the revised manuscript, Figure 7). Thanks for the reviewer gave us a chance to explain our viewpoints in detail.
In this study, TEBAC-HKUST-1 MOR exhibited excellent adsorption performance towards Pt(CN)42-, Co(CN)63-, Cu(CN)32-, and Fe(CN)63- compared with AC and as-prepared HKUST-1, indicating that the quaternary ammonium group plays a key role in the removal of metal cyanide complexes (Fig. 7). However, TEBAC-HKUST-1 MOR did not exhibited selective adsorption for Pt(II). On the other hand, both HKUST-1 and Ac also showed no selectivity for Pt(II). Fig. 7 experiment results can also be showed in Table S1. As shown in Table S1, adsorption mole number for the metal cyanide complexes by To TEBAC-HKUST-1 MOR followed the order: Fe(1.97 mmol/g) > Co (1.72 mmol/g> Pt (1.49 mmol/g) > Cu (1.38 mmol/g). To HKUST-1, maximal adsorption mole number for the metal cyanide complexes followed the order: Fe(0.63 mmol/g) > Co (0.57 mmol/g> Pt(0.39 mmol/g) >Cu (0.25 mmol/g). To AC, maximal adsorption mole number for the metal cyanide complexes followed the order: Fe(0.43 mmol/g) > Co (0.34 mmol/g> Pt(0.24 mmol/g) > Cu (0.22 mmol/g). As can be observed from Fig. 7 and Table S1, none of TEBAC-HKUST-1 MOR, HKUST-1, or AC exhibited selective adsorption for Pt(II).
Furthermore, the adsorbed Pt(CN)42- could be selectively recovered by two-step elution. First, the loaded Co(CN)63-, Cu(CN)32-, and Fe(CN)63- on TEBAC-HKUST-1 MOR could be eluted using a NaCl solution. Subsequently, NH4SCN solution was used to elute the loaded Pt(CN)42-. (Please see original manuscript).
3) Pt(CN)4 was recovered from resin upon treatment with SCN-. After that how is the resin regenerated?
Answer:
On original manuscript (line 288), We had stated the method of TEBAC-HKUST-1 MOR recovery. “The adsorbent can be regenerated by washing with a saturated sodium chloride solution.”
4) In the recycling experiments, the recovery gradually decreased. Authors should comment about it.
Answer:
We have added the PXRD of recovered MOR (Fig. 2). As can be observed in Fig. 2, after five adsorption-desorption cycles at pH=8.0, the intensities of diffraction peaks of TEBAC-HKUST-1 MOR slightly decreased, indicating that the MOR framework was only partially decomposed during the recycling experiments. On the other hand, the elution rate was less than 100%, part of active sites in the MOR was occupied by no elution samples. Hence, the recovery rates of Pt(II) gradually decreased in the recycling experiments. As shown in Figure 11, the recovery rates of Pt(CN)42- for all five cycles were over 92.0% in the mixture. The TEBAC-HKUST-1 MOR absorbent exhibited a good stability and reusability.
All revision has been highlighted in red in the manuscript. All authors agree the submission of this revised manuscript. If there is anything concerning this manuscript, please feel free to contact us. We are pleased to revise our manuscript just according to your suggestion.
Many thanks again for your reconsideration on our manuscript.
With kind regards,
Zhangjie Huang
Please see the attachment

Reviewer 2 Report
This manuscript deals with development of recovery system for platinum cyanide using functionalized metal organic resin, which is a novel piece of environmental science. Although I think it is acceptable for publication in Molecules, however, several issues should be addressed before that.
1) In this work, ion-exchange resin, and its functionalized form were used, however, no structure was shown anywhere. So, I cannot understand what happens in the resin. Authors should provide structure of resin, and how it was modified.
2) Authors explained the higher selectivity of Pt by different affinity with quartenary ammonium ion. However, resins that was not grafted with ammonium ion, also exhibited selectivity for Pt. Why?
3) Pt(CN)4 was recovered from resin upon treatment with SCN-. After that how is the resin regenerated?
4) In the recycling experiments, the recovery gradually decreased. Authors should comment about it.
Author Response
27-July-2019
Manuscript ID: molecules-557656
Dear Editors,
Many thanks for the comments on our manuscript, which is so helpful for us to make the manuscript much clearer. We are pleased to answer these questions. The manuscript has been extensively revised according to the editor and the reviewer’s comments. We really appreciate you for your suggestions. We have modified the manuscript accordingly, and detailed corrections are listed below point by point:
According to Reviewer #1’s comment, the revisions were made as follows:
1) The PXRD of recovered MOR needs to be included as well.
Answer:
Thanks for the benefit advice from the reviewer. According to the comment, we have added the PXRD of recovered MOR. After five adsorption-desorption cycles at pH=8.0, the intensities of diffraction peaks of TEBAC-HKUST-1 MOR slightly decreased, indicating that the crystallinity of MOR was only partial loss. The TEBAC-HKUST-1 MOR adsorbent exhibited a good stability and reusability. ( Please see the revised manuscript, Figure 2)
2) The authors should try to quantify TEBAC functional groups crafted in TEBAC-HKUST-1. A simple digestion of the MOR followed by HNMR should be able to provide the information. The loading of TEBAC within the MOR can be further correlated to the absorbed metal cyanide to determine whether all active sites have captured metal cyanide.
Answer:
Thanks for the benefit advice from the reviewer. To date, quantitative calculation of PSM on the MOR through 1H NMR for the one-step synthesis has been widely reported in literature (without complex intermediates) [1-4]. In this study, TEBAC-HKUST-1 MOR main are used in extraction metal cyanide complexes. Hence, the purity of the product is not main goal. Due to grafting reaction from HKUST-1 to TEBAC-HKUST-1 MOR contain three steps, implying the products are complex, containing the raw material and complex intermediates. Hence, purification of TEBAC-HKUST-1 MOR digesting solutions is difficult. The intermediate product will appear vibration peaks in 1HNMR, which serious interference quantify TEBAC functional groups crafted in TEBAC-HKUST-1. Hence, Post-synthetic modification of MOR using multistep functionalization is usually not characterized with 1H NMR[5-7]. The degree of modification usually was calculated based on results of elemental analysis[8]. In this revised manuscript, the content of nitrogen in TEBAC-HKUST-1 MOR was determined through elemental analysis (1.5 wt %.). Based on the elemental analysis result, the grafting rate of active group can be further calculated [8, 9].
As shown in Figure 7, TEBAC-HKUST-1 MOR exhibited excellent adsorption performance towards Pt(CN)42-, Co(CN)63-, Cu(CN)32-, and Fe(CN)63- compared with AC and as-prepared HKUST-1, indicating that the quaternary ammonium group plays a key role in the removal of metal cyanide complexes. From equation (1), we can deduce that the main mechanism for the adsorption of metal cyanide complexes from aqueous solutions followed an ion-exchange reaction. Quaternary ammonium groups provided large amount of active sites, which anionic on quaternary ammonium groups can exchange with Pt(CN)42-, Co(CN)63-, Cu(CN)32-, and Fe(CN)63-. The detailed adsorption mechanism and whether all active sites have captured metal cyanide need to be further studied.
References:
1. Wang, Z.Q.; M. Cohen, S. Postsynthetic Covalent Modification of a Neutral Metal-Organic Framework. J. Am. Chem. Soc. 2007, 129, 12368-12369.
2. Cohen, S.M.Postsynthetic Methods for the Functionalization of Metal Organic Frameworks. Chem. Rev. 2012, 112, 970–1000.
3. Luan, Y.; Zheng, N.N.; Qi, Y.; Yu, J.; Wang, G. Development of a SO3H-Functionalized UiO- 66 Metal–Organic Framework by Postsynthetic Modification and Studies of Its Catalytic Activities. Eur. J. Inorg. Chem. 2014, 26, 4268–4272.
4. Molavi, H.; Eskandari, A.; Shojaei, A.; Mousavi, S.A. Enhancing CO2/N2 adsorption selectivity via post-synthetic modification of NH2-UiO-66(Zr). Microporous and Mesoporous Materials, 2018, 257, 193-201.
5. Chen, C.; Wu, Z. W.; Que, Y. G.; Li, B. X.; Guo, Q. R.; Li, Z.; Wang, L.; Wan, H.; Guan, G. F. Immobilization of a thiol-functionalized ionic liquid onto HKUST-1 through thiol compounds as the chemical bridge. Rsc. Adv. 2016, 6, 54119-54128.
6. Abednatanzi,S.; Abbasi,A.; Masteri-Farahani,M. Post-synthetic modification of nanoporous Cu3(BTC)2 metal-organic. Journal of Molecular Catalysis A: Chemical, 2015, 399,10-17.
7. Haque,E.;Lee,J.E.; Jang,I.T.; Hwang, Y.K.; Chang,J.S. Adsorptive removal of methyl orange from aqueous solution with metal-organic frameworks, porous chromium-benzenedi carboxylates. Journal of Hazardous Materials, 2010, 181,535-542.
8. Chen,J.Z.; Li,K.G.; Chen,L.M.; Liu,R.L.; Huang,X.; Ye.D.Q. Conversion of fructose into 5-hydroxymethyl furfural catalyzed by recyclable sulfonic acid-functionalized metal–organic frameworks. Green Chemistry, 2014,16,2490-2499.
9. Goesten,M.G.; Juan-Alcaniz,J.; Ramos-Fernandez,E.V, et al. Sulfation of metal–organic frameworks: Opportunities for acid catalysis and proton conductivity. Journal of Catalysis ,2011, 281,177-187.
3) Scheme 1 needs more work to well present the PSM on the MOR. The authors could cut a unit cell and show the details of the binding between TEBAC and the SBU of HKUST-1. A better figure can help authors quickly understand the material.
Answer:
We are very sorry that we did not make Scheme 1 clear in the manuscript. According to the comment, we re-state Scheme 1 ( Please see the revised manuscript, Scheme 1).
In this study, a novel functional ion-exchange/adsorption metal organic resin (TEBAC-HKUST-1 MOR) was prepared by post-synthetic modification strategy. First, Cu2+ ions of HKUST-1 were chelated with the S atom of ethanedithiol. Ethanedithiol was used as the grafting agent to introduce thiol groups onto HKUST-1[1,2]. Second, the 4-vinylbenzyl chloride was immobilized onto SH-HKUST-1 by the reaction of vinyl and thiol. Finally[1], quaternary ammonium functional groups were grafted onto the MOF matrix through quaternization reaction[3].
References:
1. Chen, C.; Wu, Z. W.; Que, Y. G.; Li, B. X.; Guo, Q. R.; Li, Z.; Wang, L.; Wan, H.; Guan, G. F. Immobilization of a thiol-functionalized ionic liquid onto HKUST-1 through thiol compounds as the chemical bridge. Rsc. Adv. 2016, 6, 54119-54128.
2. Ke, F.; Qiu, L. G.; Yuan, Y. P.; Peng, F. M.; Jiang, X.; Xie, A. J.; Shen, Y. H.; Zhu, J. F. Thiol-functionalization of metal-organic framework by a facile coordination-based postsynthetic strategy and enhanced removal of Hg2+ from water. J. Hazard. Mater. 2011, 196, 36-43.
3. Wang, M. L.; Hsieh, Y. M. Kinetic study of dichlorocyclopropanation of 4-vinyl- 1-cyclohexene by a novel multisite phase transfer catalyst. J. Mol. Catal. A- Chem. 2004, 210, 59-68.
According to Reviewer #2’s comment, the revisions were made as follows:
1) In this work, ion-exchange resin, and its functionalized form were used, however, no structure was shown anywhere. So, I cannot understand what happens in the resin. Authors should provide structure of resin, and how it was modified.
Answer:
Thanks for the benefit advice from the reviewer. According to the comment, we re-state Scheme 1. ( Please see the revised manuscript).Post-synthetic modification (PSM) refers to the chemical modification of MOR after the formation of crystals, while the basic frameworks remain unchanged. In this study, HKUST-1 is selected as the carrier. Considering the strong coordination ability of metal centers (Cu2+), ethanedithiol is used as the grafting agent to introduce thiol groups. By the reaction of vinyl and thiol, 4-vinylbenzyl chloride can then be immobilized onto HKUST-1, Finally, Quaternary ammonium was grafted onto the MOR for constructing a new adsorbent for the highly effective removal of metal cyanide complexes from aqueous solutions. From Equation (1), we can deduce that the main mechanism for the adsorption of metal cyanide complexes from aqueous solutions followed an ion-exchange reaction.
To date, structural characterization of the chemical modification of MOR through 1H NMR for the one-step synthesis has been widely reported in literature (without complex intermediates) [1-4]. Due to modification reaction from HKUST-1 to TEBAC- HKUST-1 MOR contain three steps, implying the products are complex, containing the raw material and complex intermediates. Hence, purification of the MOR digesting solutions is difficult. The intermediate product will appear vibration peaks in 1HNMR, which serious interference the molecular structural characterization of the sample. Usually, the structure and property of multistep functionalization MOR are systematically characterized by the elemental analysis, FT-IR, XPS, TG , SEM, and N2 adsorption-desorption[5-7] . Furthermore, thiol-functionalization of HKUST-1(the first step)[5,8], the reaction between vinyl and thiol(the second step)[5], and quaterisation reaction(the third step)[9] have been extensively reported in the literatures. Guan et al. used ethanedithiol as grafting agent to introduce thiol groups on HKUST-1. By thiol compounds as the chemical bridge, the ionic liquid then be immobilized onto HKUST-1[5]. Qiu and co-workers illustration of the thiol-functionalization of HKUST-1 through coordination bonding between thiol and Cu2+ in MOR[8]. Quaterisation reaction between triethylamine and benzyl chloride had been reported by Wang and co-workers [9]. Hence, each step synthesis can be contrasted with the literature. The structure and property of the sample have been systematically characterized by following:
(1) The elemental analysis
According to the comment, we have added elemental analysis based on the literature method [10,11]. The elemental analysis results showed that nitrogen element content in TEBAC-HKUST-1 MOR was 1.5 wt %, The elemental analysis result further confirmed the successful grafting of the quaternary ammonium group in the TEBAC-HKUST-1 MOR. ( Please see the revised manuscript).
(2) FTIR spectra
The characteristic vibrational band of C–S group appeared at 686 cm−1, indicating that thiol groups were successfully introduced into HKUST-1 matrix [8]. Furthermore, compared with those observed for the bare HKUST-1, some new peaks were observed at 2973, 2916, and 2849 cm−1, corresponding to the C–H stretching of alkyl groups [12], indicating that quaternary ammonium was successfully grafted onto SH-HKUST-1 framework through quaternization reaction (Figure 1b). ( Please see original manuscript).
(3) XRD spectra
Compared with HKUST-1, the main characteristic diffraction peaks of TEBAC-HKUST-1 showed slight differences with those of HKUST-1, confirming that ligand functionalization does not change the original crystal structure of sample [8]. After five adsorption-desorption cycles at pH=8.0, the intensities of diffraction peaks of TEBAC-HKUST-1 MOR slightly decreased, indicating that the crystallinity of MOR was only partial loss. The TEBAC-HKUST-1 MOR adsorbent exhibited a good stability and reusability. ( Please see the revised manuscript).
(4) SEM analysis
Compared with that observed for bare HKUST-1, TEBAC-HKUST-1 MOR and the recovered TEBAC- HKUST-1 MOR maintained the same octahedron structure [5], indicating that HKUST-1 functionalization and adsorption reaction did not significantly affect the HKUST-1 structure, consistent with the XRD analyses. ( Please see original manuscript).
(5) TGA
For the TEBAC-HKUST-1 MOR, the weight loss can also be divided into three stages. The percent of weight loss of HKUST-1 from 40 °C to 200 °C was more than TEBAC-HKUST-1 MOR, because a part of the sites for water in HKUST-1 framework was replaced with the grafted quaternary ammonium. Second, the weight loss was due to the decomposition of immobilized quaternary ammonium (200 °C to 285 °C) [13] ( Please see original manuscript).
(6) N2 adsorption–desorption isotherms
Compared with as-prepared HKUST-1, both the BET surface area and total pore volume of TEBAC-HKUST-1 MOR decreased significantly. This is because the pores of as-prepared HKUST-1 framework were partially occupied by functionalized groups[8]. This also indicates that quaternary ammonium was successfully immobilized onto the MOF framework. ( Please see original manuscript)
(6) XPS
Figure 6b shows the sulfur 2p XPS spectrum of TEBAC-HKUST-1 MOR sample. The S 2p peak was resolved into three components; the binding energy (BE) components were observed at 164.8, 163.4, and 162.0 eV, corresponding to H–S, C–S, and Cu–S bonds, respectively [14, 15]. Figure 6c shows that only one peak appeared at 401.9 eV in the N 1s XPS high-resolution spectra of TEBAC-HKUST-1 MOR, consistent with those for previously reported quaternary ammonium [16]. This indicates that quaternization reaction occurred, and quaternary ammonium functional groups were grafted onto the MOFs. ( Please see original manuscript)
Hence, the structure and property of TEBAC-HKUST-1 MOR have been systematically characterized by the elemental analysis, FT-IR, XPS, TG , SEM, and N2 adsorption-desorption[5-11].
References:
1. Wang, Z.Q.; M. Cohen, S. Postsynthetic Covalent Modification of a Neutral Metal-Organic Framework. J. Am. Chem. Soc. 2007, 129, 12368-12369.
2. Cohen, S.M.Postsynthetic Methods for the Functionalization of Metal Organic Frameworks. Chem. Rev. 2012, 112, 970–1000.
3. Luan, Y.; Zheng, N.N.; Qi, Y.; Yu, J.; Wang, G. Development of a SO3H-Functionalized UiO- 66 Metal–Organic Framework by Postsynthetic Modification and Studies of Its Catalytic Activities. Eur. J. Inorg. Chem. 2014, 26, 4268–4272.
4. Molavi, H.; Eskandari, A.; Shojaei, A.; Mousavi, S.A. Enhancing CO2/N2
adsorption selectivity via post-synthetic modification of NH2-UiO-66(Zr).
Microporous and Mesoporous Materials, 2018, 257, 193-201.
5. Chen, C.; Wu, Z. W.; Que, Y. G.; Li, B. X.; Guo, Q. R.; Li, Z.; Wang, L.; Wan, H.; Guan, G. F. Immobilization of a thiol-functionalized ionic liquid onto HKUST-1 through thiol compounds as the chemical bridge. Rsc. Adv. 2016, 6, 54119-54128.
6. Abednatanzi,S.; Abbasi,A.; Masteri-Farahani,M. Post-synthetic modification of nanoporous Cu3(BTC)2 metal-organic. Journal of Molecular Catalysis A: Chemical, 2015, 399,10-17.
7. Haque, E.; Lee,J.E.; Jang,I.T.; Hwang, Y.K.; Chang,J.S. Adsorptive removal of
methyl orange from aqueous solution with metal-organic frameworks, porous
chromium-benzenedi carboxylates. J. Hazard. Mater. 2010, 181, 535- 542.
8. Ke, F.; Qiu, L. G.; Yuan, Y. P.; Peng, F. M.; Jiang, X.; Xie, A. J.; Shen, Y. H.; Zhu, J. F. Thiol-functionalization of metal-organic framework by a facile coordination-based postsynthetic strategy and enhanced removal of Hg2+ from water. J. Hazard. Mater. 2011, 196, 36-43.
9. Wang, M. L.; Hsieh, Y. M. Kinetic study of dichlorocyclopropanation of 4-vinyl- 1-cyclohexene by a novel multisite phase transfer catalyst. J. Mol. Catal. A- Chem. 2004, 210, 59-68.
10. Chen,J.Z.; Li,K.G.; Chen,L.M.; Liu,R.L.; Huang,X.; Ye.D.Q. Conversion of fructose into 5-hydroxymethyl furfural catalyzed by recyclable sulfonic acid-functionalized metal–organic frameworks. Green Chem. 2014,16, 2490-2499.
11. Goesten,M.G.; Juan-Alcaniz,J.; Ramos-Fernandez,E.V, et al. Sulfation of metal–organic frameworks: Opportunities for acid catalysis and proton conductivity. Journal of Catalysis, 2011, 281,177-187.
12. Alemdar, A.; Atici, O.; Güngör, N. The influence of cationic surfactants on rheological properties of bentonite-water systems. Mater Lett. 2000, 43, 57-61.
13. Abid, H. R.; Pham, G. H.; Ang, H. M.; Tade, M. O.; Wang, S. B. Adsorption of CH4 and CO2 on Zr-metal organic frameworks. J. Colloid Inter. Sci. 2012, 366, 120-124.
14. Laiho, T.; Leiro, J. A.; Heinonen, M. H.; Mattila, S. S.; Lukkari, J. Photoelectron spectroscopy study of irradiation damage and metal-sulfur bonds of thiol on silver and copper surfaces. J. Electron Spectrosc. 2005, 142, 105-112.
15. Muench, F.; Fuchs, A.; Mankel, E.; Rauber, M.; Lauterbach, S.; Kleebe, H. J.; Ensinger, W. Synthesis of nanoparticle/ligand composite thin films by sequential ligand self assembly and surface complex reduction. J. Colloid Interf. Sci. 2013, 389, 23-30.
16. Cao, W.; Wang, Z. Q.; Zeng, Q. L.; Shen, C. H. 13C NMR and XPS characterization of anion adsorbent with quaternary ammonium groups prepared from rice straw, corn stalk and sugarcane bagasse. Appl. Surf. Sci. 2016, 389, 404-410.
2) Authors explained the higher selectivity of Pt by different affinity with quartenary ammonium ion. However, resins that was not grafted with ammonium ion, also exhibited selectivity for Pt. Why?
Answer:
We are very sorry that we did not state clearly about Fig.7. We added Y-axis units of Fig.7(mg/g)( Please see the revised manuscript, Figure 7). Thanks for the reviewer gave us a chance to explain our viewpoints in detail.
In this study, TEBAC-HKUST-1 MOR exhibited excellent adsorption performance towards Pt(CN)42-, Co(CN)63-, Cu(CN)32-, and Fe(CN)63- compared with AC and as-prepared HKUST-1, indicating that the quaternary ammonium group plays a key role in the removal of metal cyanide complexes (Fig. 7). However, TEBAC-HKUST-1 MOR did not exhibited selective adsorption for Pt(II). On the other hand, both HKUST-1 and Ac also showed no selectivity for Pt(II). Fig. 7 experiment results can also be showed in Table S1. As shown in Table S1, adsorption mole number for the metal cyanide complexes by To TEBAC-HKUST-1 MOR followed the order: Fe(1.97 mmol/g) > Co (1.72 mmol/g> Pt (1.49 mmol/g) > Cu (1.38 mmol/g). To HKUST-1, maximal adsorption mole number for the metal cyanide complexes followed the order: Fe(0.63 mmol/g) > Co (0.57 mmol/g> Pt(0.39 mmol/g) >Cu (0.25 mmol/g). To AC, maximal adsorption mole number for the metal cyanide complexes followed the order: Fe(0.43 mmol/g) > Co (0.34 mmol/g> Pt(0.24 mmol/g) > Cu (0.22 mmol/g). As can be observed from Fig. 7 and Table S1, none of TEBAC-HKUST-1 MOR, HKUST-1, or AC exhibited selective adsorption for Pt(II).
Furthermore, the adsorbed Pt(CN)42- could be selectively recovered by two-step elution. First, the loaded Co(CN)63-, Cu(CN)32-, and Fe(CN)63- on TEBAC-HKUST-1 MOR could be eluted using a NaCl solution. Subsequently, NH4SCN solution was used to elute the loaded Pt(CN)42-. (Please see original manuscript).
3) Pt(CN)4 was recovered from resin upon treatment with SCN-. After that how is the resin regenerated?
Answer:
On original manuscript (line 288), We had stated the method of TEBAC-HKUST-1 MOR recovery. “The adsorbent can be regenerated by washing with a saturated sodium chloride solution.”
4) In the recycling experiments, the recovery gradually decreased. Authors should comment about it.
Answer:
We have added the PXRD of recovered MOR (Fig. 2). As can be observed in Fig. 2, after five adsorption-desorption cycles at pH=8.0, the intensities of diffraction peaks of TEBAC-HKUST-1 MOR slightly decreased, indicating that the MOR framework was only partially decomposed during the recycling experiments. On the other hand, the elution rate was less than 100%, part of active sites in the MOR was occupied by no elution samples. Hence, the recovery rates of Pt(II) gradually decreased in the recycling experiments. As shown in Figure 11, the recovery rates of Pt(CN)42- for all five cycles were over 92.0% in the mixture. The TEBAC-HKUST-1 MOR absorbent exhibited a good stability and reusability.
All revision has been highlighted in red in the manuscript. All authors agree the submission of this revised manuscript. If there is anything concerning this manuscript, please feel free to contact us. We are pleased to revise our manuscript just according to your suggestion.
Many thanks again for your reconsideration on our manuscript.
With kind regards,
Zhangjie Huang
Please see the attachment

Round 2
Reviewer 2 Report
I think the manuscript has been improved according to the reviewers' comments. So, it is acceptable for publication in Molecules.